# Global Climate Change Effects on Soil Microbial Biomass Stoichiometry in Alpine Ecosystems

Luyun Chen and Yongheng Gao *

Chengdu Institute of Biology, Chinese Academy of Sciences, Chengdu 610041, China
* Correspondence: gaoyh@cib.cas.cn; Tel.: +86-28-82890981

**Abstract:** Alpine ecosystems are sensitive to global climate change-factors, which directly or indirectly affect the soil microbial biomass stoichiometry. In this paper, we have compared the soil microbial biomass stoichiometry ratios of alpine ecosystems using the global average values. In the comparison, the responses and mechanisms of soil microbial biomass stoichiometry to nitrogen deposition, altered precipitation, warming, and elevated atmospheric carbon dioxide ($CO_2$) concentration in the alpine ecosystem were considered. The alpine ecosystem has a higher soil microbial-biomass-carbon-to-nitrogen ratio (MBC:MBN) than the global average. In contrast, the soil microbial-biomass-nitrogen-to-phosphorus (MBN:MBP) and carbon-to-phosphorus ratios (MBC:MBP) varied considerably in different types of alpine ecosystems. When compared with the global average values of these ratios, no uniform pattern was found. In response to the increase in nitrogen (N) deposition, on the one hand, microbes will adopt strategies to regulate extracellular enzyme synthesis and excrete excess elements to maintain stoichiometric balance. On the other hand, microbes may also alter their stoichiometry by storing excess N in their bodies to adapt to the increased N in the environment. Thus, a decrease in MBC:MBN and an increase in MBN:MBP are observed. In addition, N deposition directly and indirectly affects the soil fungal-to-bacterial ratio (F:B), which in turn changes the soil microbial biomass stoichiometry. For warming, there is no clear pattern in the response of soil microbial biomass stoichiometry in alpine ecosystems. The results show diverse decreasing, increasing, and unchanging patterns. Under reduced precipitation, microbial communities in alpine ecosystems typically shift to a fungal dominance. The latter community supports a greater carbon-to-nitrogen ratio (C:N) and thus an increased soil MBC:MBN. However, increased precipitation enhances N effectiveness and exacerbates the leaching of dissolved organic carbon (DOC) and phosphorus (P) from alpine ecosystem soils. As a result, a decrease in the soil MBC:MBN and an increase in the soil MBN:MBP are evident. Elevated atmospheric $CO_2$ usually has little effect on the soil MBC:MBN in alpine ecosystems, mainly because of two reasons. These are: (i) N is the main limiting factor in alpine ecosystems, and (ii) alpine ecosystems accumulate higher soil organic carbon (SOC) and microbes and preferentially decompose "old" carbon (C) stocks. The response of soil microbial stoichiometry to global climate change factors in alpine ecosystems is diverse, and the impact pathways are complex. Future studies need to focus on the combined effects of multiple global climate change factors on microbial stoichiometry and the mechanism of microbial stoichiometric balance.

**Keywords:** alpine ecosystem; ecological stoichiometry; microbial biomass; climate change

## 1. Introduction

Alpine ecosystems are the colder regions distributed over the globe, are situated at high altitudes and latitudes, and include various habitats such as soil, bare rock, permafrost, glaciers, and snow [1]. All these habitats are mainly characterized by low air and soil temperatures, high winds, and short growing seasons, and the soil serves as a crucial nutrient reservoir [2]. Due to their unique climate and harsh environment, alpine ecosystems are considered as amplifiers of the response to global climate change and can be considered as highly sensitive [3]. High-resolution model projections show that the

western part of the Pindus Mountains in Greece will be warmer than most of the country by 1.5 °C over the next 30 years [4,5]. The Tibetan Plateau, located in Asia, has become warmer and wetter over the last few decades [6]. By the end of the 21st century, the average annual temperature and precipitation are projected to increase by 2.8–4.9 °C and 15–21%, respectively [7]. The Arctic region has encountered an average temperature increase at a rate almost twice the global average over the past 100 years [8]. This has led to severe permafrost melting [9]. Nitrogen deposition in the Rocky Mountains of Colorado, USA, has exceeded the critical load on its environment [10]. Under global climate changes, the nutrients [11], water content, temperature [12], pH [13], and quality and quantity of plant apoplasts in alpine ecosystem soils may change [14]. All these changes might produce a direct or indirect effect on the community structure of microbes [15] and their extracellular enzymes [16].

Microbes are an essential component of soils, regulating core ecosystem processes such as organic matter decomposition, soil carbon (C) sequestration, and nutrient cycling [17]. Because microbes play a crucial role in the biogeochemical cycle, the energy and material flow of microbial-based detrital food webs is limited by C, nitrogen (N), and phosphorus (P) [18]. The cycles of C, N, and P are often coupled with each other [19], so the ratio of microbial quantities in terms of C, N, and P (C:N:P) has received considerable attention. Ecological stoichiometry is a powerful tool for studying ecosystem function [20], with its 'resource allocation theory (RAT)'. The theory proposes that microbes limited by a single element will increase the synthesis of extracellular enzymes corresponding to that element, thus breaking down and acquiring more of the limiting elements [21]. Another fundamental theory, 'the consumer-driven nutrient cycling (CNR)', states that the balance of C:N:P among the resource and the consumer and the efficiency of elemental use by consumers directly affect the flow of C, N, and P. This happens through the ecosystem, with microbes retaining their constituent elements, maintaining and metabolizing the limiting elements they need, and excreting the excess elements [18]. This shows that microbial biomass C:N:P determines the direction of its activity (nutrient sequestration or mineralization) and influences soil nutrient effectiveness. In addition, the changes in the soil microbial biomass stoichiometry resulting from the substrate addition can reflect the relative nutrient limitation of microbes [22,23]. This alters the competition between the microbes and plants and affects the nutrient supply to the plants [24]. The indication of limiting nutrients by microbial biomass stoichiometry can be targeted to increase the input of ecosystem-limiting elements, thereby improving the ecosystem sustainability. Thus, it is important to understand the response and mechanism of the stoichiometry of soil microbial biomass in the background of global changes in C:N:P stoichiometry in order to assess and predict ecosystem nutrient cycling and maintain ecosystem sustainability.

In recent years, many field experiments have been conducted in the alpine ecosystems to study the effects of climate change on the soil microbial biomass stoichiometry. However, the response of the microbial stoichiometry is very uncertain. For example, the soil microbial-biomass-carbon-to-nitrogen ratio (MBC:MBN), in response to the nitrogen enrichment, was found to be reduced [25,26], or a non-significant effect was found [27], and the effect of warming showed varying results with an increase [28], decrease [29], or no change [30]. The different results may be due to the different microbial stoichiometry strategies, the microbial community change direction, and the plant competition intensity. Thus, it is necessary to conduct a comprehensive overview of the alpine ecosystems to reveal the generalizable patterns and regulatory factors of the soil microbial biomass stoichiometry in response to climate change. This can increase our ability to predict the future climate change impacts on ecosystems. In recent years, several meta-analyses have been conducted to reveal the general pattern of warming on the MBC:MBN at a global scale [31]. Additionally, the individual and cumulative effects of increasing the three climate change factors (nitrogen deposition, warming, and carbon dioxide ($CO_2$)) on the microbial biomass stoichiometry were studied [32]. All these studies have improved our understanding of the ecosystem stoichiometry balance and the microbial response to the climate change

factors at a global scale. However, they have neglected the response of extremely sensitive alpine ecosystems, as a complete ecosystem type, to climate change. Therefore, an overview of the response of the alpine ecosystem microbial stoichiometry to climate change is still lacking, and the general response of the alpine soil microbial stoichiometry to climate change remains unclear. In the present study, we review the existing research literature to compare the differences between the soil microbial stoichiometry and the global averages of factors in alpine ecosystems. Another goal was to analyze the responses and mechanisms of soil microbial stoichiometry in alpine ecosystems in relation to nitrogen deposition, warming, precipitation change, and $CO_2$ increase. We were also devoted to analyzing the shortcomings of the current study and proposing scientific issues that need to be addressed in this field. All of this would provide a scientific basis for the in-depth research of biogeochemical cycles in alpine ecosystems. Since the global climate change issue happens to be a recently developed threat, studies related to it are important for the accurate assessment of the response of alpine ecological processes.

## 2. Stoichiometry Characteristics of Soil Microbial Biomass in Alpine Ecosystems

The microbial biomass stoichiometry of alpine ecosystems differs from the global average because of the special climatic conditions and vegetation types (Table 1). The MBC:MBN of alpine ecosystems is higher than the global average, probably due to the following reasons: (i) alpine ecosystems have harsher climatic conditions such as low temperatures, high altitudes, and nutrient deprivation. Plants produce more developed root systems for up taking better nutrients [33], and faster root turnover and the production of root exudates will provide more nutrients to microbes, resulting in increased microbial biomass [29,34]. However, in alpine ecosystems, an N-deprived region, a well-developed plant root system, also gives plants more of an ability to compete with microbes for N [33,35]. This makes the soil microbes of alpine regions more strongly N-limited than those in other regions. So, alpine ecosystems have a relatively higher MBC:MBN. (ii) The altered soil microbial community structure also affects soil microbial stoichiometry [36]. Since alpine ecosystems have lower temperatures and harsher climatic conditions than other regions, fungi are more tolerant and are adapted to survive in this ecosystem [37,38]. Fungi have a higher carbon-to-nitrogen ratio (C:N) than bacteria [17], and, thus, a higher fungi:bacteria (F:B) ratio tends to represent a higher MBC:MBN. (iii) Bacteria adjust their investment strategy depending on the environment in which they live. As a result, in alpine ecosystems, bacteria invest more in cell structures, such as cell walls, to resist extreme conditions, e.g., low temperatures, and thus have a higher MBC:MBN [39].

**Table 1.** Summary of C:N:P stoichiometric ratios for soil microbial biomass in alpine ecosystems and global biomes.

| Region | Ecosystem Type | MBC:MBN | MBC:MBP | MBN:MBP | MBC:MBN:MBP | Reference |
|---|---|---|---|---|---|---|
| Tibet Plateau | alpine meadow | 14.53 | 117.02 | 8.13 | 118:8:1 | [40] |
| | | 10.23 | 48.0 | 4.68 | 47.9:4.68:1 | [41] |
| | | 12.78 | | | | [42] |
| | alpine wetland | 50.56 | 184.89 | 5.42 | 275:5:1 | [40] |
| | alpine steppe | 13.49 | 80.0 | 6.03 | 81.3:6.03:1 | [41] |
| Polar | Polygonal tundra | | | 9.7 | | [43] |
| | low arctic tundra | 11.8 | 22.93 | 1.97 | | [44] |
| Subarctic | tundra | 14.4 | | | | [45] |
| The Rocky Mountains | tundra | 8.3 | | | | [46] |
| Global average | | 7.6 | 42.4 | 5.6 | 42:6:1 | [47] |
| | | 8.6 | 59.5 | 6.9 | 60:7:1 | [22] |

In the Tibetan Plateau, the soil microbial-biomass-carbon-to-phosphorus ratio (MBC:MBP) is higher than the global average, which correlates with the relative nutrient limitation of its soils. The C:N and carbon-to-phosphorus (C:P) ratios in soils in the Tibetan plateau are significantly lower than the global average [41,48]. This indicates that soil microbes are relatively more C-limited and are expected to have a higher carbon use efficiency (CUE) [17]. Thus, more C will be stored in soil microbial biomass, resulting in alpine ecosystems with a larger MBC:MBP than the global average.

In the polar regions, there are significant differences in MBC:MBP and MBN:MBP among the different alpine types. In the Arctic polygonal tundra, MBN:MBP is significantly higher than the global average. It is also known that the soil N content in the Arctic polygonal tundra is much greater than the soil P content, and a large proportion of soil P (40%) is stored within the microbial biomass. However, only a tiny proportion of N (5%) is stored within the microbes [43]. This suggests that, in the polygonal tundra ecosystems, P limitation is more severe, and, therefore, there is a higher MBN:MBP. In contrast, the MBC:MBP and MBN:MBP in the Low Arctic tundra are significantly lower than the global averages. On the other hand, Low Arctic tundra soils have very low nitrate ($NO_3^-$) contents [44], and, in the Alaskan tundra region, where environmental conditions are similar to those of the Low Arctic, microbes are strongly N-limited. This condition has therefore been limiting microbial growth and soil organic carbon (SOC) decomposition [49]. The ultimate result is an increased C limitation. The Lower Arctic tundra may have the same nutrient-limited situation and thus have lower MBC:MBP MBN:MBP ratios.

## 3. Impact of Global Change on Soil Microbial Biomass Stoichiometry in Alpine Ecosystems

### 3.1. Impact of Increased N Deposition

Atmospheric nitrogen (N) deposition has been enhanced by 3–5 times over the past century by the application of fertilizer N and fossil fuel burning [50]. It is estimated that, by 2050, global N deposition will be almost doubled compared to that in the early 1990s [51,52]. Additional N entering into the soil can directly affect soil microbial stoichiometry by affecting the nutrient supply balance and soil properties [26,53]. It can also indirectly affect soil microbial stoichiometry by altering the nutrient redistribution and plant community composition within the above-ground vegetation [54,55]. At the same time, microbes will respond to nutrient changes in the substrate by adjusting their stoichiometric characteristics, community structure, extracellular enzyme ratios, and element utilization efficiency (Figure 1) [17,56].

Several studies in alpine meadows have revealed no significant effects of N addition on soil microbial C:N [57,58]. When nutrient ratios in the substrate environment change, soil microbes can optimize resource allocation and regulate the efficiency of element utilization to maintain stoichiometric balance [17,56]. Extracellular enzymes are the mediators and tools for the microbial decomposition of organic matter. They help to break down macromolecular organic matter, including plant cell wall polymers, e.g., cellulose, hemicellulose, and lignin, into smaller soluble molecules. This facilitates bioabsorption [37], and when a single element limits microbes, they increase the synthesis of the extracellular enzyme corresponding to that element to maintain their stoichiometric balance [21]. One study in alpine meadows confirmed that N addition increased β-1,4-glucosidase activity in the soil and led to the upregulation of soil microbial genes. It helped in the degradation of chitin, cellulose, hemicellulose, and lignin to enhance C acquisition in order to balance the higher N source utilization [54]. Furthermore, CNR states that microbes retain their constituent elements, maintaining and metabolizing the limiting elements they need and excreting the excess elements [18]. Studies in alpine meadows showed that N addition increased the abundance of denitrifying bacteria [54], which significantly increased the rate of $N_2O$ emissions. As a result, reduced soil respiration [58] occurred, suggesting that N addition may cause microbes to adjust their element utilization and retention strategies and maintain their stoichiometric balance.

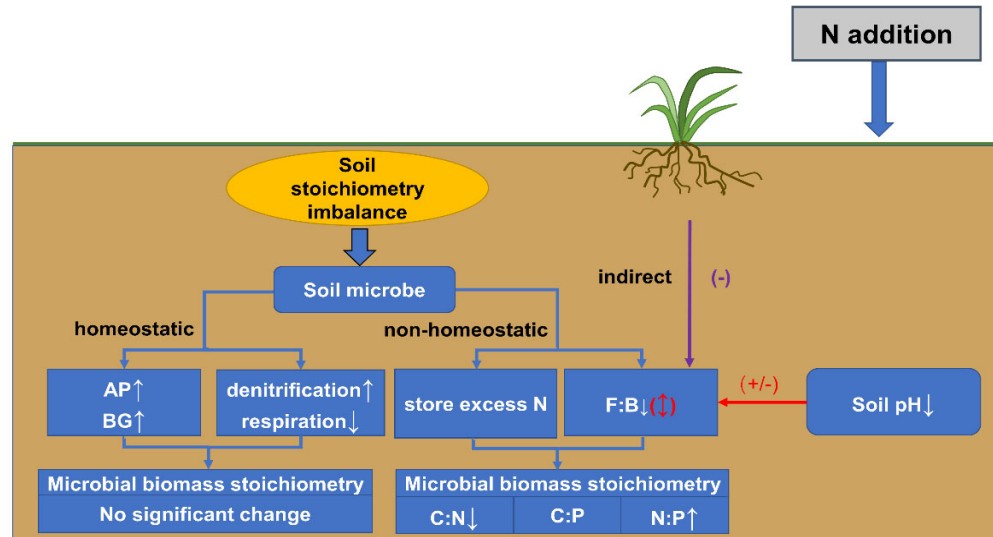

**Figure 1.** A conceptual framework for the responses of soil microbial biomass stoichiometry to N addition in alpine ecosystems. ↑, increase; ↓, decrease; (↕), increase or decrease in response to soil pH decline; (−), negative effect; (+/−), negative or passive effect; AP, acid phosphatase; BG, β-1,4-glucosidase; F:B, fungi-to-bacteria ratio.

Some studies carried out in the alpine ecosystems have also shown that N addition decreases MBC:MBN, reflecting the plasticity of microbial biomass stoichiometry [18]. A meta-analysis in the Qinghai-Tibet Plateau revealed a more significant increase in MBN than in MBC under N deposition conditions [59]. Studies in the Arctic Tundra also showed that soil microbes can store excess N in the short term [46]. All these findings suggest that microbes adapt to substrate stoichiometry imbalances by storing excess nutrients [17]. In addition, changing the community structure or increasing the abundance of specific strains is an important way for microorganisms to adapt to stoichiometric imbalances [37,58]. Fungal and bacterial growth rates and carbon use efficiency differ [60,61], with fungi being better suited to live in low-quality substrates (high C:N). At the same time, bacteria have been observed to be more abundant in substrates with low C:N ratios [36]. Furthermore, N addition alters the nutrient balance of the soil substrate, decreasing soil C:N and thus reducing the fungal abundance or increasing the bacterial abundance of the microbial community and decreasing the F:B [27,62–64].

Soil microbial community composition is also susceptible to changes in soil properties induced by N deposition [65]. Additional N input to the soil increases the $H^+$ concentration released during soil nitrification and ammonium N biosorption [13]. The process therefore decreases soil pH. Several studies have reported that soil pH has been negatively corelated with bacterial abundance and positively correlated with fungal abundance [26,65]. In all these kinds of relationships, bacteria generally exhibit lower C:N ratios than fungi [66]. Thus, a decrease in F:B leads to a reduction in MBC:MBN [17]. However, some studies have also shown that N addition can negatively affect bacteria more than fungi [67]. Compared to bacteria, fungi usually have a wider pH optimum [68] and a higher tolerance to calcium and magnesium ions leached by soil acidification. So, N addition may also increase the F:B [69]. In this case, the change in the microbial community composition due to the adaptation to the substrate nutrient balance is opposite to the shift in the microbial community composition due to the change in soil properties. How this ultimately affects microbial stoichiometry needs to be further investigated.

N addition affects the microbial biomass stoichiometry ratio through changes in microbial communities. Some studies have found that excess inorganic N added into the soil reacts with soil carbon in a condensation manner. This creates a condition where it is difficult to decompose compounds, e.g., lignin, alkyl-, and aromatic C [70,71]. It also inhibits the

activity of enzymes that break down complex C, such as lignin-degrading enzymes [72,73], thus reducing C utilization by soil microbes and decreasing the MBC:MBN ratio.

Additional N input also affects plants and indirectly alters microbial community structure, thus affecting soil microbial biomass stoichiometry. Firstly, N addition alters plant nutrient redistribution and changes plant community composition [74]. Several studies in alpine meadows have found that N addition leads to a decrease in the C:N of plant leaves, roots, and shoots and a decline in legumes [26,27], whereas plant litter and root secretions are nutrient sources on which microbes depend, and a high-N-containing nutrient supply is more favorable for bacterial survival [75]. Secondly, N addition changes the plant's nutrient investment strategy. It reduces plant investment in arbuscular mycorrhizal fungi (AMF), which causes a decrease in AMF abundance, resulting in a lower F:B [31,76]. This phenomenon thus causes a reduction in the MBC:MBN. However, other studies have proved that the effects of N addition on microbial communities were not correlated with changes in vegetation [77]. N addition directly affected microbial community changes rather than feeding back to microbial communities indirectly through changes in vegetation composition. For example, a study by Pietikäine [78] in a subarctic meadow showed that the symbiosis of AMF with plants was not influenced by N addition. The variability that occurred in these studies suggests that whether N addition affects microbial stoichiometry indirectly through plants is uncertain and may vary depending on environmental factors or experimental conditions.

As an important component of microbes, P is essential for maintaining normal microbial growth and genetic metabolism. Similar to the secretion of more C-degrading enzymes by microbes to maintain stoichiometric balance, the N:P imbalance of the substrate after N addition also induces microbes to secrete more P-degrading enzymes. In alpine ecosystems, N addition increases Acid phosphatase activity in the soil [79,80]. There is also higher phosphorus acquisition enzyme activity in C-richer rhizosphere soil but more β-1,4-glucosidase (BG) in P-rich bulk soil [24]. This suggests that microbes can selectively increase the secretion of the corresponding hydrolytic enzymes of limiting elements to maintain stoichiometric balance.

Microbes not only secrete the corresponding enzymes to obtain the relatively lacking nutrients in order to maintain their stoichiometric balance but also adopt non-homeostatic strategies to change their stoichiometry to adapt to environmental changes. A study by Liu [26] found that N addition increased MBN:MBP and decreased MBC:MBN but had no effect on MBC:MBP. All this suggests that the stoichiometry of microbial biomass after N addition changes was mediated by N elements only, reflecting the plasticity of microbial biomass stoichiometry.

### 3.2. Impact of Global Warming

The Intergovernmental Panel on Climate Change (IPCC) reported that, by 2017, anthropogenic global warming was about 1 °C above pre-industrial levels [81] and had predicted that the global surface temperature would increase by 1.1–4.8 °C by the end of the century [82]. Alpine ecosystems have a harsh climate, often limited by low temperatures. As one of the most sensitive ecosystems in the world to global warming, their temperature rise is more significant compared to other ecosystems [37]. So, the structure and function of the soil microbial community, the fungal and bacterial abundance, and the enzyme activity of those ecosystems are more sensitive to temperature increases [83,84]. Thus, their microbial biomass stoichiometry is also responsive to global warming (Figure 2).

Some studies in alpine regions have shown that rising temperatures increase MBC and MBN but decrease MBC:MBN [25,29], which is consistent with the results of the Meta-analysis on a global scale [31]. In alpine ecosystems, microbes are limited by low temperatures [85] and can adapt to this factor at the cellular level by reducing metabolic activity or dormancy. Thus, temperature can increase the competitive advantage of microbes [83], directly and positively affecting microbial biomass C and N [86]. In addition to this, warming increases plant biomass in alpine zones, which in turn increases plant

litter and fine root turnover. As a result, more root exudates are produced by the root system, providing more available C and N sources for soil microbes [87] and thus increasing MBC and MBN. The positive response of MBN to warming is more significant than that of MBC. Warming causes a greater increase in net N mineralization and nitrification [84,88], which increases inorganic N in the soil [89] and improves soil N supply, which is positively correlated with MBN [90].

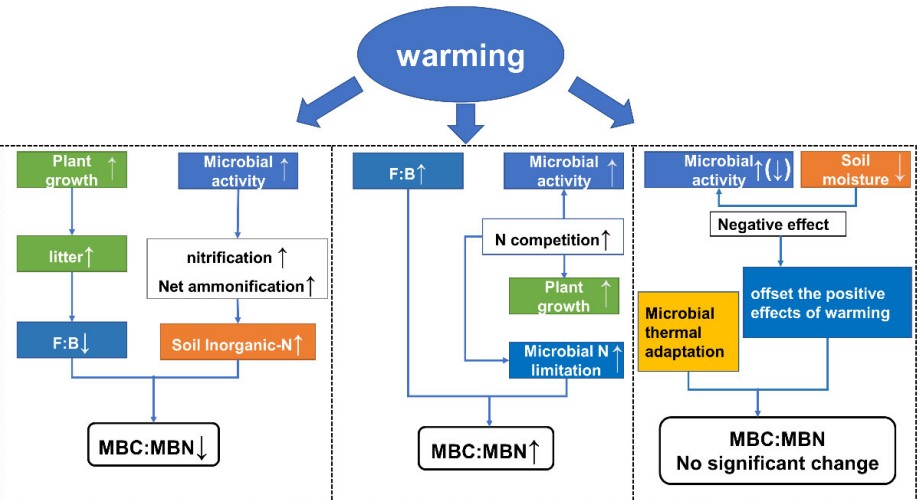

**Figure 2.** A conceptual framework for the responses of soil microbial biomass C:N to warming in alpine ecosystems. ↑, increase; ↓, decrease; F:B, fungi-to-bacteria ratio.

In addition, the increase in plant growth due to climate warming will greatly contribute to the formation of soil microbial community structure, changing the ratio of fungi to bacteria and causing changes in microbial abundance and stoichiometry [91]. In alpine shrub ecosystems, warming treatments caused a significant decrease in the F:B [29], and this phenomenon was similarly observed in warming experiments in subarctic heath ecosystems [92]. This may be because of the increase in plant products (i.e., root exudates and fresh litters) in these alpine ecosystems, which may directly benefit the actinomycetes and bacterial communities rather than the fungal community [29,93]. Warming-induced increases in plant growth may also enhance the competition between plants and soil microbes for labile soil nutrients, thereby indirectly increasing the relative abundance of oligotrophic taxa such as actinomycetes [94].

Some studies in alpine regions have also showed that warming increases MBC:MBN, mainly due to plant-microbial competition for N in alpine ecosystems. Alpine regions tend to be generally N-limited [95], and warming increases the amount of available N in the soil and plant growth and also increases plant competition for N [96]. Microbial biomass and N turnover rates are much shorter than the length of the plant growing season. Therefore, microbial N may have been recycled by plants multiple times [97]. Additionally, plants are exposed to a number of opportunities to sequester recycled N, which may result in high N sequestration and retention by plants during the growing season [98]. So, microbial N in the alpine ecosystems may decrease under warming conditions. In addition, in N-limited soils, warming may lead to a microbial "nitrogen-digging" mechanism, i.e., the decomposition of hard-to-degrade organic matter to obtain more N, which in turn leads to more C for microbes, increasing MBC:MBN [25].

Fungi may also have greater abundance ratios to bacteria under warming conditions [99,100]. Meta-analyses of the cold ecosystems have shown that warming increased the biomass or abundance of subterranean fungi, fungus-feeding fungi, and plant roots without affecting the net biomass of bacteria or archaea at high northern latitudes [101]. Warming may allow fungi to promote the growth of C-rich, nutrient-seeking saprophytic

networks while inhibiting the formation of N-rich, nutrient-storing vesicles [102], resulting in a net increase in fungal biomass.

However, most studies in alpine regions show that the effect of warming on MBC:MBN is insignificant. Microbes have the ability to maintain their homeostasis and adapt to warming [103]. Long-term warming may lead to the thermal adaptation of microbes. There are three mechanisms to explain microbial adaptation under warming: (i) growth at a specific temperature brings phenotypic advantages without any genotypic change, (ii) genotypic adaptation within species (evolution), (iii) species classification: species that are already genetically better adapted to a specific temperature regime will compete for other less adapted species [104]. These cases may not change the microbial biomass under warming conditions [105]. Temperature increases can also have an impact on soil factors, thus negatively affecting microbes indirectly and offsetting the positive effect of temperature increases on soil microbial biomass in alpine ecosystems. Increasing temperature reduces the water content of the soil [106,107], causing soil drying, inhibiting microbial physiological performance, and reducing the diffusion of substrates to microbes. All these functions lower the microbial activity [108], in addition to the fact that, under warming, readily decomposable substrates are consumed more rapidly [97,109,110]. Thus, microbes are required to invest more energy in decomposing recalcitrant substrates but invest less energy in growth [111]. In addition, microbial maintenance costs increase at higher temperatures, reducing the proportion of energy obtained for growth [112]. Some studies have found no effect of long- and short-term temperature increases on plant biomass, soil temperature, and moisture [89,113]. Vegetation, soil moisture, and temperature are crucial factors affecting the composition of microbial activity [89]. So, the adaptation of vegetation and soil to temperature also leads to insignificant changes in microbial biomass stoichiometry.

The effects of warming on MBC:MBP and MBN:MBP have been relatively little studied in alpine regions, and there exists a greater scope for the exploration of their responses and mechanisms. Existing sporadic research studies of long-term warming experiments in the subarctic tundra have found that warming significantly increased soil orthophosphate ($PO_4$-P) content. Parallelly, decreased acid phosphatase activity, increased enzyme C:N, and significantly decreased MBP result in an increase in the MBN:MBP [114]. Similar findings were made in long-term warming experiments in the Arctic tundra [115], where warming led to a decrease in phosphodiesterase activity, which, in turn, led to a decrease in MBP, and the authors speculated that warming possibly enhanced microbial N or P limitation to limit enzyme production [116]. However, a 21-year warming experiment on the Tibetan Plateau has revealed that warming did not affect hydrolase activity and ratios, nor did it significantly affect MBC:MBP. This suggests that it may be due to the thermal adaptation of microbes as a result of long-term warming [103]. For these different results, the mechanisms need to be further explored in the future.

*3.3. Impacts of Changing Precipitation Patterns*

The global precipitation pattern will be altered by climate warming, and the frequency of extreme drought and wetness periods is expected to increase [8,117]. Changes in precipitation will directly impact microbial communities by affecting soil moisture conditions [118], as well as by altering the availability of soil substrates [86], which, in turn, will affect soil microbial biomass stoichiometry (Figure 3).

Drought simulation experiments in the Qinghai-Tibet Plateau region showed that drought increased MBC:MBN [119], which is consistent with the results of meta-analyses at the global scale [120]. Under drought conditions, the microbial community shifts to fungal dominance [121]. Fungi have a filamentous structures and can reach and utilize substrate nutrients even under low soil moisture and diffusivity [122], whereas bacteria require water films for motility and substrate diffusion [123]. Moreover, fungi have chitinous cell walls that are more resistant and resilient to changes in moisture [66]. Thus, fungi have a higher drought tolerance than bacteria. In addition, to balance the decrease in soil water potential under drought conditions, fungi accumulate polyols that are high in C without N

as solutes in their bodies [124], which could also explain the increase in MBC:MBN under drought conditions.

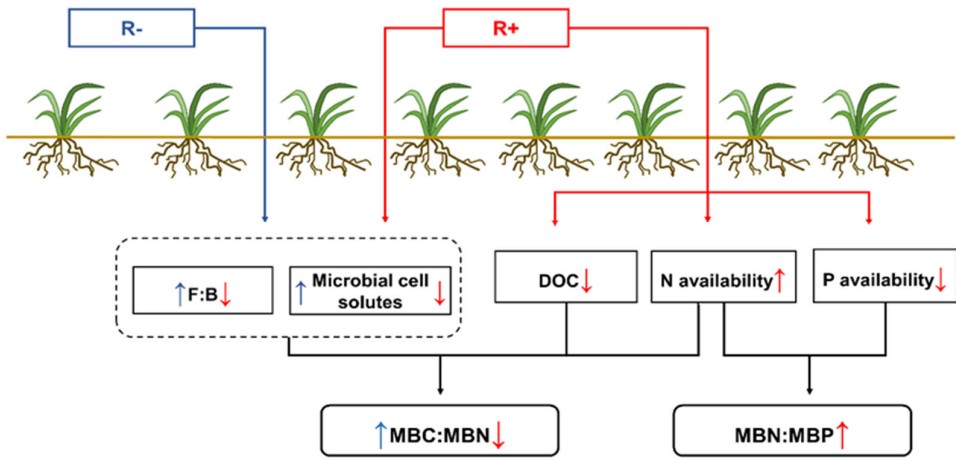

**Figure 3.** A conceptual framework for the responses of soil microbial biomass stoichiometry to precipitation changes in alpine ecosystems. ↑, increase; ↓, decrease; R−, decreased precipitation; R+, increased precipitation; F:B, fungi-to-bacteria ratio; DOC, dissolved organic carbon. The blue lines represent the influencing pathway of decreased precipitation; the red lines represent the influencing pathway of increased precipitation.

In contrast to the drought situation, increased precipitation was found to decrease MBC:MBN in water addition experiments [125], and field observation experiments similarly found that MBC:MBN decreased with increasing soil water contents [42]. The reasons for this may be the following: first, higher precipitation has a positive effect on bacterial biomass [126]. Second, with the rise in soil moisture, the content of available N in the soil increases, alleviating the nutrient limitation of soil microbes in alpine ecosystems [125] and therefore leading to a higher MBN. Third, the soil dissolved organic carbon (DOC) content in alpine ecosystems decreases significantly with increasing rainfall frequency [127]. DOC depends on the input of soluble plant residues, and the leaching of soil DOC content will become more severe with increasing rainfall frequency [42,127], leading to a decrease in DOC content, an increase in microbial C limitation, and a reduction in MBC. Finally, if precipitation increases after a drought and the soil rewets, microbes will release through respiration, aggregation, or cell membrane transport through the drought-accumulated solutes to prevent cell rupture caused by the inflow of soil water into the cells due to osmotic pressure differences [128]. This process will release large amounts of $CO_2$ and DOC [124,129], resulting in a decrease in MBC.

Increased precipitation also heightens MBN:MBP in alpine grasslands and meadows [130], which is consistent with studies on a global scale [131]. Precipitation can enhance P leaching, reducing P availability and increasing microbial P limitation. In the absence of P availability, microbes will replace phospholipids with other P-free lipids, reducing the cellular demand for P [132] and leading to a decrease in MBP.

### 3.4. Impact of Elevated Atmospheric $CO_2$ Concentration

During the period 1750–2019, $CO_2$ concentrations increased by 131.6 ppm (47.3%), and global atmospheric carbon dioxide ($CO_2$) concentrations are now higher than at any time in the last two million years, at least [133]. Due to the high $CO_2$ concentrations in soil pores, the effect of $CO_2$ on soil microbes is more likely to be through changes in the quantity and quality of plant biomass and plant root production and exudation rather than through the direct effects of surface $CO_2$ [134,135].

Elevated $CO_2$ stimulates plant photosynthesis [136], which increases the net primary productivity of plants. At the same time, a portion of the additional fixed C is allocated to the root system and stimulates the growth of mycorrhizal fungi [137]. A higher C:N ratio in

the soil also favors the survival of fungal communities [36]. Some non-alpine studies found that elevated $CO_2$ increased fungal abundance but had little effect on bacteria, leading to an increase in F:B [131,134], and, thus, an increase in MBC:MBN occurs. However, in a 9-year experimental study on $CO_2$ enrichment in an alpine forest line, elevated $CO_2$ was found to affect neither microbial communities nor the change in MBC:MBN [138]. Alpine ecosystems are more N-limited and low temperature-limited for plants and microbes than other ecosystems [95]. So, even if elevated atmospheric $CO_2$ concentrations increase the C supply, the effect on the plant aboveground part and the fine root biomass and microbial biomass remains small. This was confirmed by a study in alpine grasslands, where the researchers found that elevated $CO_2$ alone did not affect microbial biomass C, N, and, obviously, MBC:MBN. When nutrient limitation was removed (e.g., through wet N deposition), microbes would utilize the additional carbon introduced into the soil system [139].

In addition, it has been shown that elevated $CO_2$ increases the accumulation of non-structural carbohydrates and the synthesis of C-based secondary metabolic products (e.g., lignin) in plants [137]. Therefore, inevitably, an increase in the C:N and C:P ratios of plant tissues [140] occurs, which indirectly affects the microbial community structure. However, alpine ecosystems accumulate high SOC contents over time because of low-temperature limitations [141]. Only a relatively small fraction of new C is imported into the soil by plants each year, and only a small fraction of microbes (some fungi) metabolize new plant-derived C [138], so the increase in plant tissue C ratios due to elevated $CO_2$ concentrations also has a minor impact on soil microbial biomass stoichiometry in alpine ecosystems.

Elevated $CO_2$ also leads to a decrease in plant stomatal conductance and an increase in water use efficiency, reducing plant transpiration and thus increasing soil moisture [142]. Under wetter soils, gaseous N losses occur via nitrification/denitrification, or an increase in $NO_3^-$ loss through leaching occurs [143]. All these phenomena reduce the relative availability of N, possibly leading to a decrease in MBN:MBP. Data accumulated via research on the response of soil microbial biomass stoichiometry to elevated $CO_2$ in alpine ecosystems are relatively sparse and need to be further explored in the future.

## 4. Future Perspectives

Microbial biomass stoichiometry is a powerful tool with a profound ability to investigate the intrinsic mechanism of ecosystem nutrient cycling and the linkages among ecosystem components. However, there are still many questions and directions to be addressed in studying the response patterns and mechanisms of soil microbial biomass stoichiometry in alpine ecosystems in the context of global climate change.

(1) Global climate change is often a combination of multiple factors. Different global change drivers may have cumulative or opposite effects on the environment and microbes, e.g., elevated $CO_2$ increases soil moisture, but warming decreases soil water. Thus, future studies should use integrated experiments to investigate how different climate change drivers affect alpine ecosystem stoichiometry to build better models and to explore the microbial stoichiometric response under natural conditions.

(2) Microbes respond to changes in substrate stoichiometry with two completely different strategies: (i) maintaining stoichiometric homeostasis and (ii) maintaining non-stoichiometric homeostasis. However, the processes and mechanisms involved are complex and are currently not well understood. In the future, a combination of experimental approaches should be adopted to further explore the mechanisms of microbial communities in response to changes in substrate stoichiometry at different experimental scales, from macroscopic to microscopic (e.g., molecular studies of sugars, amino acids, proteins, RNA) features.

(3) P is a critical limiting factor in nature, and elevated $CO_2$ and increased N deposition alleviate C and N limitation in ecosystems to some extent, while possibly leading to increased MBC:MBP and MBN:MBP. Current research on microbial biomass stoichiometry in alpine ecosystems is mainly focused on MBC:MBN. In future studies, research

on microbial biomass stoichiometry related to microbial P should be increased to explore the effects of P limitation on alpine microbial biomass stoichiometry in the context of global climate change factors.

(4) Plant and microbial interdependence and competition in alpine ecosystems. In the context of global change, plants and microbes respond individually, while there exists a strong interaction between the two. However, in the current study, it is relatively rare to consider plant-litter-soil as a complete system. To address the above issues, the linkage with plant and litter stoichiometry can be explored under the premise of studying soil microbial stoichiometry to better understand the nutrient cycling in alpine ecosystems in the context of global climate change factors.

(5) In alpine ecosystems that respond rapidly to global climate change, applying microbial stoichiometry models can extend the available data to explore the role of microbial stoichiometry in biogeochemical cycling in this rapidly changing habitat. Currently, there is no universal chemometric model suitable for alpine ecosystems, and the role of chemometrics in biogeochemical models should be emphasized in the future, thus enabling the development and improvement of coupled plant-soil-microbial stoichiometry models for alpine ecosystems.

**Author Contributions:** Conceptualization, Y.G; methodology, L.C.; writing—original draft preparation, L.C.; writing—review and editing, L.C. and Y.G.; supervision, Y.G.; funding acquisition, Y.G. All authors have read and agreed to the published version of the manuscript.

**Funding:** This research was funded by the Key Research and Development Program of Sichuan Province of China (grant no. 2022YFS0489) and the International Communication and Cooperation Project of Qinghai Province (grant no. 2019-HZ-807).

**Institutional Review Board Statement:** Not applicable.

**Informed Consent Statement:** Not applicable.

**Conflicts of Interest:** The authors declare no conflict of interest.

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
