# Peer review of "Global Climate Change Effects on Soil Microbial Biomass Stoichiometry in Alpine Ecosystems"

_land, doi:10.3390/land11101661_

Round 1

Reviewer 1 Report

It is presented a comprehensive review study of recent knowledge the global climate change effects on soil microbial biomass stoichiometry in alpine ecosystem in  the context of global climate change factors. In this study, the authors compared the soil microbial biomass stoichiometry ratio of alpine ecosystems using the global average values. Usually, the alpine ecosystem has higher soil microbial biomass carbon to nitrogen ratio (MBC:MBN) than the global average. It was summarised that for warming, there is no clear pattern in the response of soil microbial biomass stoichiometry in alpine ecosystems, thereby elevated atmospheric CO2 usually has little effect on soil MBC:MBN in alpine ecosystem. However, more sophisticated future studies need to focus on the combined effects of multiple global change factors on microbial stoichiometry and the mechanism of microbial stoichiometric balance.

The structure of paper is well organized, and the review study brings new findings, interesting information, and knowledge.

The manuscript contains no objective discrepancies nor mistakes of formal character. Attention deserves only a denomination “AMF” in the text of manuscript (see Lines 212 and 218. What author mean with this abbreviation AMF?

Author Response

亲爱的审稿人,

非常感谢您抽出宝贵时间审阅此稿件。我非常感谢您的所有意见和建议!请在下面找到我的逐项回复,并在重新提交的文件中找到我的修订/更正。

1)在手稿文本中只应注意“AMF”的面额(见第212和218行。作者对这个缩写AMF是什么意思?

答:非常感谢您的批准和如此仔细的检查。AMF“是指丛枝菌根真菌,我们在第229-230行添加了缩写”AMF“的定义。

再次感谢您的仔细评论和积极建议!

Reviewer 2 Report

The article deals with an interesting topic like climate change. Particularly, the article examines the climate change effects on soil microbial biomass stoichiometry in alpine ecosystem.

Line 44. Moreover, in areas with complex terrain, such as forest and alpine ecosystems the model’s future projections present uncertainties and resolution play a key role in reliable estimations (Stefanidis, 2021; Tolika et al.,2016)

Stefanidis, S. (2021). Ability of Different Spatial Resolution Regional Climate Model to Simulate Air Temperature in a Forest Ecosystem of Central Greece. J. Environ. Prot. Ecol, 22, 1488-1495.

Tolika, K., Anagnostopoulou, C., Velikou, K., & Vagenas, C. (2016). A comparison of the updated very high resolution model RegCM3_10km with the previous version RegCM3_25km over the complex terrain of Greece: present and future projections. Theoretical and applied climatology, 126(3), 715-726.

Add some comments about the importance of soil microbial biomass stoichiometry for ecosystem sustainability.

In the last paragraph of the introduction, clearly state the research gap answered from the current research.

Round 2

Reviewer 2 Report

The article can is accepted for publication in the current form.